ecology, behaviour, environmental science

species composition, ecosystem function, habitat selection, herbivory processes, community structure, climate change

**Author for correspondence:**
Laura E. Richardson
e-mail: l.richardson@bangor.ac.uk

# Coral species composition drives key ecosystem function on coral reefs

Laura E. Richardson[1,2], Nicholas A. J. Graham[1,3] and Andrew S. Hoey[1]

[1]ARC Centre of Excellence for Coral Reef Studies, James Cook University, Townsville, Queensland 4811, Australia
[2]School of Ocean Sciences, Bangor University, Menai Bridge LL59 5AB, UK
[3]Lancaster Environment Centre, Lancaster University, Lancaster LA1 4YQ, UK

LER, 0000-0002-1284-4011; NAJG, 0000-0002-0304-7467; ASH, 0000-0002-4261-5594

Rapid and unprecedented ecological change threatens the functioning and stability of ecosystems. On coral reefs, global climate change and local stressors are reducing and reorganizing habitat-forming corals and associated species, with largely unknown implications for critical ecosystem functions such as herbivory. Herbivory mediates coral–algal competition, thereby facilitating ecosystem recovery following disturbance such as coral bleaching events or large storms. However, relationships between coral species composition, the distribution of herbivorous fishes and the delivery of their functional impact are not well understood. Here, we investigate how herbivorous fish assemblages and delivery of two distinct herbivory processes, grazing and browsing, differ among three taxonomically distinct, replicated coral habitats. While grazing on algal turf assemblages was insensitive to different coral configurations, browsing on the macroalga *Laurencia* cf. *obtusa* varied considerably among habitats, suggesting that different mechanisms may shape these processes. Variation in browsing among habitats was best predicted by the composition and structural complexity of benthic assemblages (in particular the cover and composition of corals, but not macroalgal cover), and was poorly reflected by visual estimates of browser biomass. Surprisingly, the lowest browsing rates were recorded in the most structurally complex habitat, with the greatest cover of coral (branching *Porites* habitat). While the mechanism for the variation in browsing is not clear, it may be related to scale-dependent effects of habitat structure on visual occlusion inhibiting foraging activity by browsing fishes, or the relative availability of alternate dietary resources. Our results suggest that maintained functionality may vary among distinct and emerging coral reef configurations due to ecological interactions between reef fishes and their environment determining habitat selection.

## 1. Introduction

Global climate change and mounting local stressors are degrading ecosystems via species extirpations and introductions, modifying the composition of assemblages and threatening ecological function [1,2]. Non-random species turnover, ordered by the susceptibility of organism traits [3], is increasing the taxonomic and functional similarity of communities [4–6]. These changes can disrupt ecosystem processes, such as habitat provisioning [7,8], primary productivity [9], trophic energy flow [10], nutrient cycling [11,12] and pollination [13]. While evidence exposes a coherent pattern of ecological change across biomes [14], variation exists from the individual to community level in how ecological structure, ecosystem processes and ongoing disturbance dynamics interact [15,16]. For effective and adaptive local management, better understanding is needed of the extent to which different, and in some cases emerging, species configurations support processes critical to ecological stability [17].

We focus on coral reefs, one of the most biodiverse but threatened ecosystems [18], to elucidate how the composition of habitat-building species (i.e. corals)

influences key ecosystem functions. Climatic changes and local human impacts have reduced populations of corals, resulting in unprecedented loss of coral cover and marked shifts in coral species composition due to differential susceptibilities of corals to thermal stress, severe storms, predation by crown-of-thorns starfish and poor water quality [19,20]. Typically stress-sensitive, topographically complex branching corals (e.g. Acroporidae) are replaced by more robust, prostrate corals (e.g. Mussidae, Poritidae) following disturbance [20,21]. The composition and cover of coral species are key determinants of the structural complexity of reef habitats [21,22], and can exert considerable influence over the taxonomic and functional structure of reef fish assemblages [6]. However, the capacity of altered coral species configurations to support key ecosystem processes despite ongoing disturbance is largely unknown and of growing concern [20,23].

Herbivory, the consumption of algal material, is dominated by fishes on coral reefs with relatively intact fish assemblages. Herbivory processes can promote coral dominance by reducing the cover and/or biomass of algae, though the amount of herbivory necessary will depend on the extent of substrate available to algae, background nutrient levels that can accelerate algal increase [24] and the effect of anthropogenic ocean warming on corals [25]. If herbivory is sufficient, it can mediate competitive interactions with corals [26], mitigate shifts to macroalgal dominance following extensive coral mortality, and facilitate recovery of coral populations [27]. However, the distribution of herbivorous fishes and their rates of herbivory can be highly spatially variable; among regions [28,29], latitudes [30], across the continental shelf [31], with the amount of nutrients entering the system [32] and among reef zones [33,34]. Importantly, rates of herbivory by fishes often vary among sites within-reef zones [35,36], with studies relating variation to differences in habitat structural complexity [28], the cover of live coral [29,37], the relative palatability of resident algal communities [34,38], predation pressure or competition for resources [39]. Where variation in herbivory is driven by the differential composition of benthic reef habitats [35], this may carry implications for the variable functioning of distinct coral species configurations. However, relationships between coral species composition and herbivory processes by fishes at the within-reef scale remain unclear.

Herbivory processes are diverse, carried out by multiple species that perform complementary, and in some cases functionally overlapping, roles in removing algae from the reef substrate [40,41]. For example, grazing fishes (including algal croppers/detritivores, scrapers and excavators) feed on surfaces covered by epilithic algal matrices (EAM: a conglomerate of algal turfs, macroalgal propagules, sediment, detritus and microbes [42]), but have limited capacity to remove large fleshy macroalgae [38]. By feeding on EAM covered surfaces, grazers maintain algal communities in a cropped state, reduce the growth of macroalgal propagules within the EAM, reduce coral–algal competition and thereby facilitate settlement, growth and survival of corals and coralline algae [41]. By contrast, macroalgal browsers typically feed on larger fleshy macroalgae and have the potential to reverse phase shifts by removing macroalgae biomass, facilitating the recovery of coral populations [27,43]. Understanding the extent to which different configurations of structurally distinct corals maintain populations of herbivorous fishes and the critical functions they provide is paramount for the management of ecological integrity yet is largely unknown.

The primary objective of this study was to investigate how grazing and browsing herbivory processes by reef fishes varied among coral habitats that differed in coral species composition and structural complexity across within-reef scales [22]. Using a combination of *in situ* surveys and transplanted algal assays across three replicated habitats characterized by the predominance of distinct coral taxa (*sensu* [22]), we specifically ask the following questions. (i) Do the structure of herbivorous fish assemblages and rates of grazing and browsing vary among reefs characterized by distinct coral habitats? (ii) What is the relative influence of coral species composition and structural complexity, and herbivore biomass on these herbivory functions within reefs?

## 2. Material and methods

### (a) Study sites

This study was conducted in April and May 2016 on coral reefs surrounding the continental high islands of the Lizard Island Group, 33 km off the mainland coast of Cape Flattery in the northern Great Barrier Reef (14°41′ S, 145°27′ E; electronic supplementary material, figure S1). Three replicate sites of three taxonomically and structurally distinct coral habitats were selected on shallow (less than 6 m) reefs, based on surveys completed in September 2015 [22]. These three habitats were characterized by predominant cover of: (i) branching *Porites* (mostly *P. cylindrica*); (ii) soft coral (mostly *Lobophyton, Sarcophyton* and *Sinularia*); and (iii) mixed coral assemblages (mostly staghorn, corymbose and plating *Acropora*, massive and branching *Porites, Lobophyton, Sarcophyton*) (electronic supplementary material, figure S2 and table S1). The study coincided with a large-scale coral bleaching event at Lizard Island [44], with fish and benthic communities affected across the study sites [6]. At each site, we quantified herbivore fish and benthic assemblages (including the extent of coral bleaching), and the consumption of algal turfs and a locally abundant macroalga. All sites (each greater than 250 m × 5 m) were positioned on the leeward side of the islands protected from the prevailing southeast swell, had comparable geomorphology and water clarity [45,46], and were separated by more than 500 m.

### (b) Benthic composition and herbivore assemblages

The benthic composition was quantified along six 30 m point-intercept transects at each site, recording the substratum immediately under the tape every 25 cm (120 points per transect). Transects were positioned approximately 2.5 m from, and parallel to, the reef-sand interface. Substratum categories were hard (scleractinian) corals identified to genus (or species where possible) and morphology, soft (alcyonacean) corals identified to genus, 'other sessile invertebrates' (mainly clams, sponges and ascidians), macroalgae identified to genus, 'dead substrata' (dead coral and pavement, covered in EAM), rubble and sand. For corals directly under surveyed points, the extent of bleaching was assessed *in situ* using the CoralWatch colour reference card estimating coral tissue colour saturation on a 6-point scale (1–2 considered 'bleached').

To account for behavioural plasticity, functional overlap and uncertainty regarding specific herbivore species particularly across their different life-history stages [40,43,47], the abundance and total length (TL; nearest centimetre) of all nominally herbivorous fishes (i.e. Acanthuridae, Kyphosidae, Pomacanthidae, Scarinae, Siganidae, Pomacentridae; electronic supplementary material, table S2) were visually censused along the same six 30 m transects used to quantify benthic assemblages. Omnivorous herbivores known to consume algae in addition to zoobenthos

and zooplankton were also censused. Fishes of more than 10 cm TL were recorded within a 5 m wide belt while initially deploying the transect tape to minimize disturbance to fish assemblages, and those less than or equal to 10 cm TL were recorded within a 1 m wide belt on the return swim. Fish abundance estimates were standardized per 150 m² and converted to biomass (kg ha$^{-1}$) using published species length–weight relationships (electronic supplementary material, table S2). All surveyed species were categorized into six nominal groups (i.e. macroalgal browsers, croppers/detritivores, scrapers, excavators, farmers and omnivorous herbivores) based on their diet and feeding behaviour (electronic supplementary material, table S1).

## (c) Rates of herbivory

To quantify rates of grazing on algal turfs among habitats, we exposed established turf algal communities on terracotta tiles (10 × 10 × 1 cm) with to resident herbivores at each site for 7 days. To establish turf algal communities, 79 tiles were deployed at a single shallow reef site (approx. 2 m depth) at Lizard Island, covered with plastic mesh (5 cm square mesh) to exclude feeding by large herbivorous fishes and left in situ for six months. After this period, the tiles were collected, and eight haphazardly selected tiles were deployed at each of the nice sites. Six tiles were exposed to local herbivores, and two were placed inside individual exclusion cages (300 × 300 × 300 mm; 12 mm² steel mesh) to determine if observed changes in algal turf height were due to herbivory at each site. An additional caged tile was included at seven of the nine sites (all sites except one mixed coral site and one soft coral site). The tiles were deployed at each site by securing to individual cement pavers with a galvanized steel nut and bolt through the centre of the tile. The pavers were placed on horizontal surfaces that were free of live coral at each site, with greater than 10 m between adjacent pavers/tiles. Exclusion cages were cleaned of fouling organisms (mostly algae) every 2 to 3 days. The initial height of the turf algal community was quantified at nine uniformly spaced points in situ using callipers (nearest millimetre) across the upper surface of the tile immediately after deployment (mean ± s.e. = 4.89 mm ± 0.13; no significant variation among habitats, lme, $F_{2,6} = 1.14$, $p = 0.38$), and again after 7 days.

To quantify rates of macroalgal browsing, transplanted 'bioassays' (hereafter 'assays') of the red macroalga Laurencia cf. obtusa were used. Laurencia was selected as it was the most abundant macroalga on reefs surrounding Lizard Island at the time of the study, and Laurencia spp are known to be consumed by herbivorous reef fishes on the Great Barrier Reef [48,49]. Thalli of Laurencia were collected by hand from a local shallow reef flat and placed in an aquarium (6000 l) with flow-through seawater within 30 min of collection. Whole thalli of similar size were spun in a salad-spinner for 30 s to remove excess water, and the wet weight recorded (to the nearest 0.1 g). The initial mass (mean ± s.e.) of each assay was 45.4 ± 1.0 g. Six haphazardly selected assays were transplanted to each site between 9.30 and 10.30, with three exposed to resident herbivore assemblages and three placed within adjacent herbivore exclusion cages (300 × 300 × 300 mm) for 24 h. Each caged assay was positioned within 2 m from its paired exposed assay, and adjacent assay pairs were separated by a minimum of 10 m. Assays were deployed with a short (less than 10 cm) length of PVC-coated wire (2 mm diameter) around the thallus base and attached to a small lead weight. Small plastic tags placed adjacent to assays were used to identify individual thalli. After 24 h assays were collected, spun and re-weighed. This procedure was replicated on three non-consecutive days at each site ($n = 9$ exposed assays per site).

To identify herbivorous fish species removing Laurencia biomass, stationary underwater video cameras (GoPro) recorded feeding activity on up to three (mean = 2.2 assays) haphazardly selected assays at each site on each day. Each camera was attached to a dive weight (2 kg) and positioned approximately 1 m from each assay, with a scale bar temporarily placed adjacent to each assay at the start of filming to allow calibration of fish sizes on video footage. Filming commenced immediately after assays were deployed and was continuous for 2.2–4.4 h (variable duration due to differences in battery life among cameras). This procedure was replicated on each day of the experiment (3 per site), resulting in 20.5 ± 1.7 h (mean ± s.e.) of video observations for each site (189 h in total). Body size (TL) and number of bites taken from the Laurencia by each species on the video footage were recorded. To account for variation in fish body size on algal mass removed per bite, mass-standardized bite impact was calculated as the product of the number of bites and the estimated body mass for each individual (following [50]). Bite impact was then standardized per hour to account for varying video lengths (mass-standardized bites per hour).

## (d) Data analysis
### (i) Benthic composition and herbivore assemblages
Variation in the total cover of hard and soft coral, bleached coral (hard and soft), macroalgae, and dead substrata and macroalgae combined among habitats was analysed with linear mixed-effects models (lme in nlme; fixed factor: habitat, random factor: site), with Tukey's multiple comparisons post hoc to identify significant differences (multcomp).

Variation in taxonomic composition of herbivorous fish assemblages among habitats was visualized with non-metric multi-dimensional scaling based on Bray–Curtis similarities of log-transformed biomass data (kg ha$^{-1}$; log $(x + 1)$ – transformed), and differences assessed with two-factor nested PERMANOVAs (9999 permutations), using habitat (fixed factor) and site (random within habitat), and Monte Carlo pairwise comparisons. Variation in total biomass (kg ha$^{-1}$) of all herbivores (log-transformed) among habitats was assessed with a linear mixed-effects model fitted with Gaussian residual structure (lme in nlme; fixed: habitat; random: site). Variation in herbivore species richness (Poisson distribution), total herbivore abundance and grazer biomass (combined biomass of croppers/detritivores, scrapers, excavators and omnivorous herbivores; both models using negative binomial distributions) was assessed with generalized mixed models to accommodate non-stable variances and alternative exponential residual distributions (glmer in lme4), followed by Tukey's multiple comparisons to identify significant differences among habitats (multcomp). Variation in macroalgal browser biomass was assessed using the same fixed and random effects, but with a zero-inflated negative binomial generalized linear mixed-effects model (glmmTMB in glmmTMB, and multiple comparisons for glmmTMB in https://cran.r-project.org/web/packages/glmmTMB/vignettes/model_evaluation.html#multcomp).

### (ii) Rates of herbivory
Variation in the reduction in height of algal turfs, and reduction in biomass of Laurencia assays among habitats was assessed with linear mixed-effects models with a Gaussian residual structure (with lme in nlme). Models included habitat, treatment (exposed versus caged-control), and their interaction (fixed effects), and site (random effects). Day of deployment was included as an additional random effect for the model of the reduction in Laurencia biomass. A generalized mixed-effects model with a negative binomial distribution was used to assess variation in feeding on Laurencia (total mass-standardized bites per hour) due to exponential residual distribution, with habitat (fixed), site and day of deployment (random). Multiple linear regression and information-theoretic model selection was used to assess the relative influence of centred site-mean environmental variables on the change in exposed assays (assay loss) where significant differences were found among habitats: the first axis of a principal components analysis of benthic composition (accounting for

61.7% variation in benthic composition of transects among habitats); per cent cover of dead substrata and macroalgae; and underwater visual census (UVC) estimated biomass of nominal herbivore groups (grazers or browsers). Total coral cover was collinear with the cover of dead substrata and macroalgae so was not included. All variables had a variance inflation factor (VIF) less than 2, and multi-model inference (including null models) estimated by ranked changes in AICc less than 2.

Model assumptions of homogeneity of variance and normality were validated with visual assessment of Pearson residuals, and multicollinearity of explanatory variables in the multiple linear regression analysis was assessed by calculating relative VIF. Where variance was heterogeneous among habitats, a constant covariance structure was fitted (i.e. change in macroalgal weight; percentage cover of macroalgae, bleached coral, and hard and soft coral). All analyses were performed in R (R Core Team 2019), and Primer v. 6 with PERMANOVA+.

## 3. Results

### (a) Benthic composition and herbivore assemblage structure

Total coral cover was significantly higher in branching *Porites* habitats than mixed coral habitats (contrast 16.6%, confidence interval (CI): 27.48 | 5.67) and intermediate in soft coral habitats (figure 1a; electronic supplementary material, table S3). There was no significant variation in the total cover of bleached coral or macroalgae among habitats (electronic supplementary material, table S3), the latter being low across all sites (mean: 0.3–1.4%) and comprised mainly of *Padina*, *Halimeda* and *Dictyota*. However, the cover of dead substrata and macroalgae (predominately turf algae) was lower in branching *Porites* than mixed coral habitats (contrast: 17.2%, CI: 3.09 | 31.36) and intermediate in soft coral habitats (electronic supplementary material, table S3).

The taxonomic composition of herbivorous fish assemblages differed significantly between branching *Porites* habitats and soft coral habitats, largely driven by differences in the relative biomass of grazing species, such as the parrotfishes *Chlorurus microrhinos*, *Scarus niger*, *S. rivulatus*, and the surgeonfishes *Acanthurus blochii* and *Ctenochaetus striatus* (PERMANOVA, pseudo-$F = 2.47$, d.f. = 2,53, $p = 0.004$, unique permutations = 280; pairwise test, $p$(MC) = 0.004; figure 2). Herbivore assemblages from the mixed coral habitat did not differ from the other two habitats. Variation in herbivore assemblages (species richness, total biomass, biomass of grazers and browsers) among habitats was inconsistent with the cover of turf and macroalgae described above. Herbivore species richness and total herbivore biomass (kg ha$^{-1}$; log-transformed) were significantly greater in mixed coral habitats than soft coral habitats, and intermediate in branching *Porites* habitats (figure 1b,c; electronic supplementary material, tables S3 and S4) as was the biomass of grazers (electronic supplementary material, table S5). Conversely, the biomass of browsers was significantly greater in branching *Porites* habitats than soft coral habitats (CI: 1.43 | 11.98), and intermediate in mixed coral habitats (figure 3; electronic supplementary material, tables S4 and S5).

### (b) Rates of herbivory

Although the reduction in height of algal turf assays differed among habitats (both caged and exposed), with the greatest reduction in the soft coral habitat and lowest reduction in

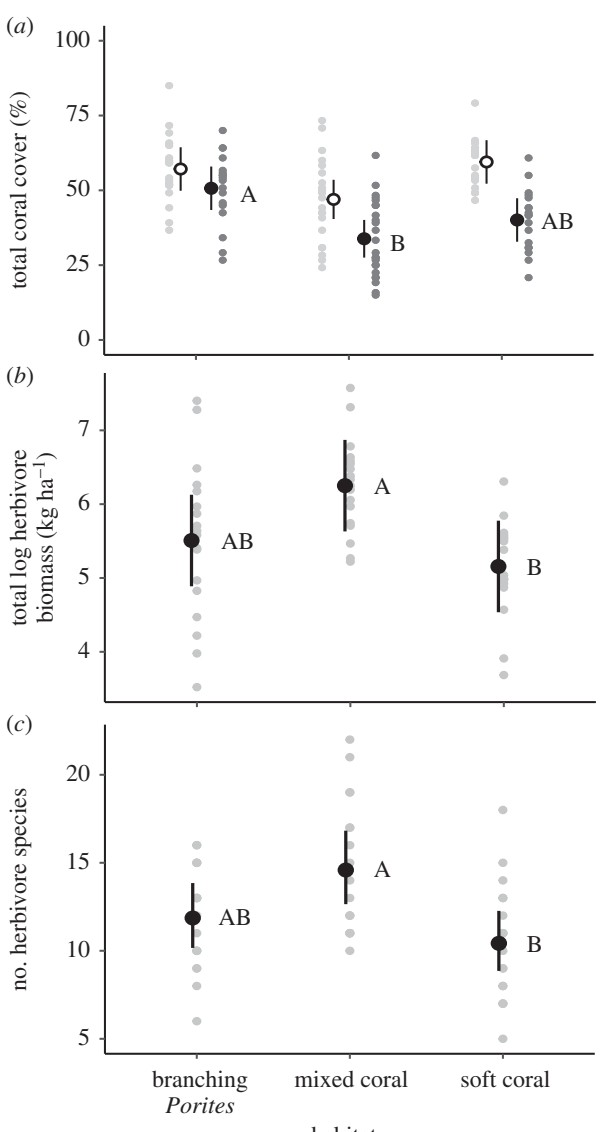

**Figure 1.** Among-habitat variation (fitted values ±95% confidence intervals) in, (a) total coral cover (hard and soft coral) in September 2015 (white; [6]), and April 2016 (black); (b) total herbivore biomass (log-transformed, kg ha$^{-1}$); (c) number of herbivore species. Partial residuals in grey; contrasting letters indicate significant differences among habitats (Tukey, $p < 0.05$).

the branching *Porites* habitat, the difference between caged and exposed tiles (i.e. the reduction in height due to herbivores) was consistent among habitats (figure 4a; electronic supplementary material, table S6). The reduction in algal turf height on tiles exposed to local herbivore assemblages was significantly greater than on caged tiles across all habitats (CI: 0.21 | 1.38).

The reduction in *Laurencia* biomass was greater in the mixed coral and soft coral habitats than in the branching *Porites* habitats where the change in weight of exposed assays did not differ significantly to caged assays (figure 4b; electronic supplementary material, table S6). Model selection of variables that explained the reduction in mass of *Laurencia* assays yielded two models within ΔAICc < 2 of the top model (electronic supplementary material, table S7). The most parsimonious included the cover of dead substrata and macroalgae (relative importance: 1.00) and first axis of the principal component of benthic composition among habitats (PC1; relative importance: 0.53), and was 1.1 times more plausible than the second-ranked model (electronic supplementary

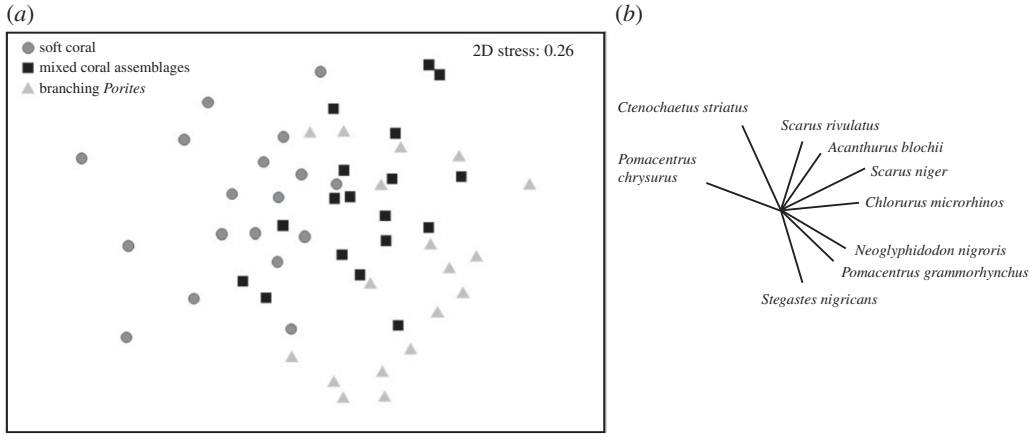

**Figure 2.** (a) Non-metric multi-dimensional scaling analysis showing variation in the taxonomic composition of herbivorous fishes among surveyed coral habitats, using transect-level log (x + 1) transformed data. (b) The relative contribution of species to the observed variation in composition (greater than 0.5 Pearson correlation).

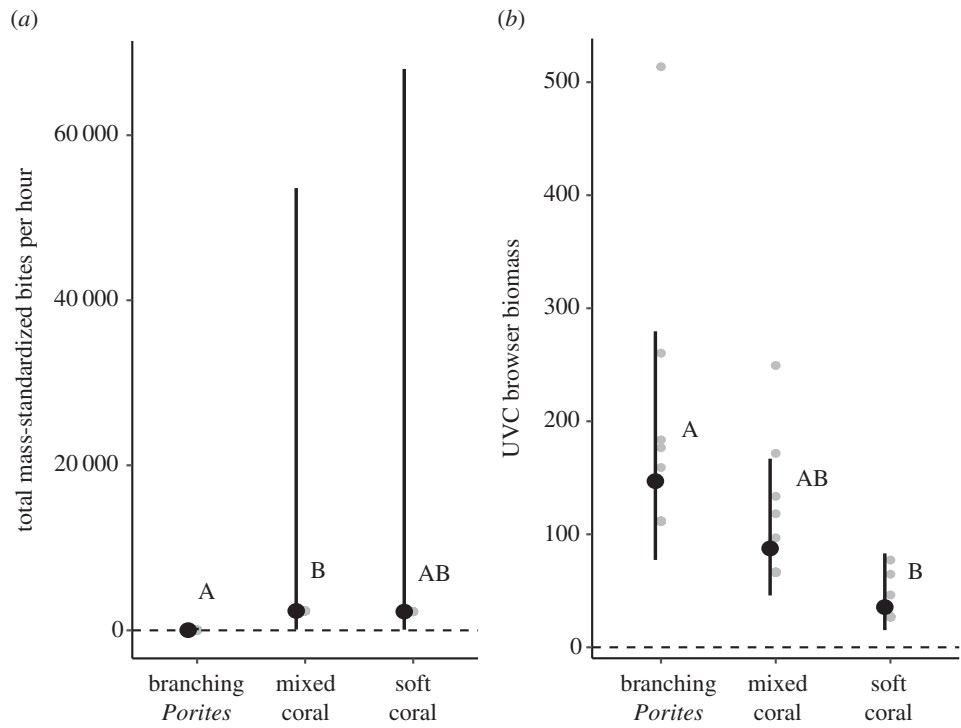

**Figure 3.** Among-habitat variation (fitted values ±95% confidence intervals) in (a) feeding rates on *Laurencia* assays by all species and (b) visual biomass estimate of all nominal browsers (kg ha$^{-1}$). Contrasting letters indicate significant differences among habitats (Tukey, $p < 0.05$).

material, table S7). Across both top models, dead substrata and macroalgae had a significant (CI: 0.04 | 0.98) and positive effect on assay weight change, while PC1 did not have a significant effect (CI: −1.76 | 0.29) (electronic supplementary material, table S7).

Total feeding on *Laurencia* assays was significantly lower in the branching *Porites* habitat than mixed coral habitat (CI: 3.32 | 3347.43), and intermediate in the soft coral habitat (figure 4; electronic supplementary material, table S6). Analysis of video footage revealed 35 species of reef fishes taking bites from exposed assays across all habitats, with four species accounting for 96% of total mass-standardized bites: *Naso brevirostris* (69%), *Siganus doliatus* (13%), *Naso vlamingii* (9%) and *Pomacanthus sexstriatus* (6%). Feeding by each of these species was highly variable among assays and sites, and poorly reflected UVC estimates of fish biomass (electronic supplementary material, figure S3). Of these four

species, only *P. sexstriatus* was recorded feeding in branching *Porites* habitats.

## 4. Discussion

Shifts in the composition of habitat-forming species and consequences for the function of ecosystems pose new challenges for conservation as the composition of assemblages that rely on habitats for food and shelter reorganize [7,51]. Focusing on coral reefs, we show that the taxonomic and functional composition of herbivorous fish assemblages, and rates of browsing, but not grazing, differed among taxonomically distinct coral habitats. Browsing on the red macroalga *Laurencia* was greatest in soft coral and mixed coral habitats, and lowest in branching *Porites* habitats. These differences in the consumption of *Laurencia* were best predicted by variation in

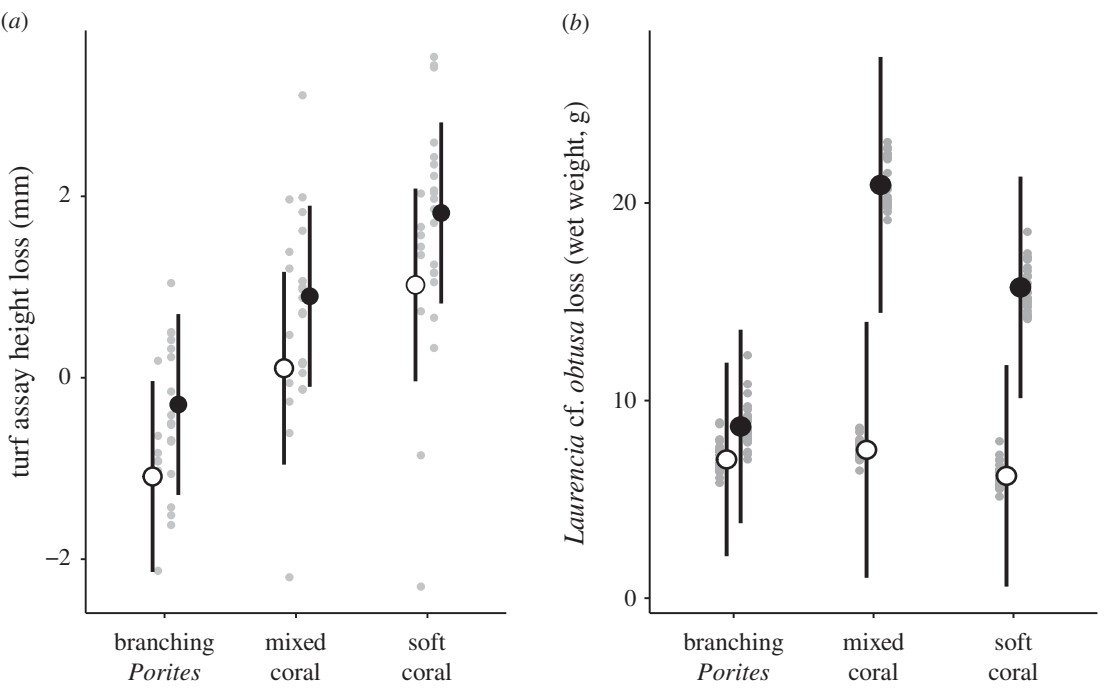

**Figure 4.** Among-habitat variation (fitted values ±95% confidence intervals) in assay loss of (*a*) turf algae (mean turf height, mm) and (*b*) *Laurencia* (wet weight, g); caged assays (white), exposed assays (black), partial residuals (grey).

both the composition and cover of benthic assemblages, with the highest rates of removal in habitats with the lowest coral cover, lowest structural complexity, and highest cover of dead substrata and macroalgae. Interestingly, rates of browsing on *Laurencia* were poorly reflected by visual estimates of the biomass of browsing fishes, despite browsing fishes being recorded in all three habitats. By contrast to browsing rates, grazing on algal turfs did not differ among habitats. This contrast highlights that different environmental mechanisms, such as those determined by the influence of differential habitat characteristics on foraging behaviour, may shape the functional impact of key species and functional groups such that shifts in species configurations under mounting disturbances may have varied consequences for maintained ecosystem function [7,8].

The observed variation in rates of browsing among habitats was best predicted by the cover and composition of benthic communities, indicating that particular habitat characteristics may influence foraging behaviour and/or habitat selection by browsing reef fishes. The cover of live coral and structural complexity of reef habitats typically have positive effects on the abundance, biomass, and diversity of herbivorous fish communities [33,52], and rates of herbivory [35,36]. By contrast, however, we found that browsing on *Laurencia* was greater in habitats with lower coral cover that had lower structural complexity, and higher cover of dead substrata and macroalgae (e.g. mostly mixed coral habitats, largely characterized by massive and branching *Porites*, *Sarcophyton*, *Lobophyton*). Conversely, while branching *Porites* habitats were the most structurally complex [22], had the highest coral cover, and the greatest observed biomass of browsing fishes among habitats, no significant reduction in *Laurencia* biomass was detected over a 24 h period. The negative relationship between the cover of structurally complex corals (and conversely the positive relationship with the cover of dead substrata and macroalgae) and browsing rates may be related to increased levels of visual occlusion during feeding

in high-relief habitats and hence greater risk of foraging [53,54]. Studies show the physical topography of structurally complex habitats can inhibit access to algal resources at fine scales (i.e. between coral branches [55]), and can alter the foraging behaviour of fishes by reducing their visual fields and thereby enhancing perceived predation risk [53]. Such findings reflect patterns of habitat use in other terrestrial and aquatic systems where foraging species favour open over structurally complex habitats due to the enhanced ability to detect approaching predators (e.g. African savannahs [56,57]; temperate intertidal rocky shores and mudflats [58]; alpine forests [59]; European grasslands [60]; temperate arable areas [61]). Indeed, evidence shows that visual obstruction can increase vigilant predator-scanning behaviour at the cost of time spent foraging in various taxa [57,60]. Moreover, perceived predation risk can also be mediated by body size with larger prey less susceptible to predation [56]. Of the four main species recorded feeding on *Laurencia* in our study, only *P. sexstriatus* was observed feeding within the structurally complex branching *Porites* habitat, despite *N. brevirostris* and *S. doliatus* being recorded in visual surveys of that habitat. *P. sexstriatus* was the largest-bodied species observed (mean biomass ± s.e.: 670 g ± 77; other species mean biomass 195–539 g), potentially reducing predation risk and enabling less discriminant foraging activity.

The positive relationship between browsing and the cover of dead substrata and macroalgae (which was highly collinear with the cover of live coral), also suggests that habitat condition may influence the foraging behaviour of herbivore fishes. Indeed, feeding rates by herbivorous reef fishes can be higher in degraded areas, of often lower topographic complexity [37]. By feeding where food resources are more abundant, animals may maximize net energy gain by reducing energetic costs of movement [62,63], and risk of predation associated with moving larger distances between resource patches [64]. In our study, differential browsing rates may relate to the differential availability of algal dietary

resources [35,39] following the bleaching event that caused coral loss and increased the cover of turf algae (figure 1*a*) [6] at our study sites (between 52.4 and 71.4% cover of dead substrata). Browsing on *Laurencia* was greatest in mixed coral habitats that also had the highest cover of dead substrata and macroalgae as a result of the bleaching (due to loss of mainly *Acropora* and soft coral taxa [6]), and highest biomass and diversity of herbivorous fish. Increased cover of algae (predominately turf communities) following large-scale bleaching-induced coral mortality and subsequent increases in the abundance and/or biomass of herbivorous fishes (e.g. [65]), has led to suggestions that herbivorous fish populations may be food limited in areas of high coral cover [66]. However, this relationship may not hold at very low levels of macroalgal cover [34], such as those observed in the present study (mean: 0.3–1.4% cover).

While visual census estimates show macroalgal browsing herbivores are present in each of the studied habitats, browser biomass was a poor predictor of browsing rates. This is consistent with previous studies of herbivorous coral reef fishes [36,50] and processes in other systems (e.g. the decomposition of dung by invertebrates [67]; pollination by bees [13]) in which abundance shows little relation to their functional impact. The discrepancy between observed browser presence and function in our study may also reflect the high mobility and opportunistic foraging behaviour of roving herbivores [68], or the diver-negative behaviours of some fishes [69]. The utility of using the density or biomass of browsing herbivores as a proxy for macroalgal removal may be further hindered by the plasticity and opportunistic diets among herbivorous fishes [47], and a potential bias in the literature classifying browsers as those species known to feed on large fleshy brown macroalgae versus those that consume other fleshy macroalgae [48].

By contrast to browsing, there were no detectable differences in grazing on the algal turf assays among habitats. This provides further evidence of a disconnect between the observed density and realized the impact of functional groups of herbivorous fishes. Despite no detectable differences in grazing rates, among-habitat differences in herbivore assemblages were largely driven by differences in the biomass of grazing species. The lack of among-habitat variation in grazing may be related to the high diversity of fishes that feed on algal-turf covered substrata [41], and their response diversity to changes in benthic composition [70]. Similarly, the lack of observed differences may be due to grazing herbivores preferentially targeting sparse and short early successional turfs and avoiding later successional dense turf assemblages [71]. Feeding rates and foraging behaviours of grazing coral reef fish species have been shown to vary with the condition and structure of reef habitats and algal communities, however, responses tend to be species specific [37]. The among-habitat variation in the changes in the turf height on caged tiles was interesting as, despite feeding by large herbivorous fishes being excluded, there was a decline in height in soft coral habitat and increase in branching *Porites* habitat which may be related to grazing by small invertebrates and/or differences in algal productivity [72]. Similarly, negative values of turf height loss for both caged and exposed assays in branching *Porites* habitats may be due to high algal productivity in that habitat, warranting further investigation.

Our results provide new evidence of the variable influence of the composition and cover of habitat-building corals on two key functions on coral reefs—grazing and browsing—based on comparisons among three taxonomically distinct coral habitats. While the use of *Laurencia* has provided valuable information on the variable browsing behaviour among habitats, previous studies have shown rates of macroalgal browsing can be dependent on the macroalgae used due to feeding preferences of local herbivore assemblages [48,49]. Therefore, further investigation using other commonly occurring macroalgae may offer insight into behavioural variation among habitats of a broader suite of herbivores. Similarly, herbivory processes can vary with depth, exposure and reef zonation [33,73,74]. Our study compared relatively small experimental assay units among habitats within in a sheltered lagoon environment. Therefore, further study across a wider range of environmental gradients, reef zones, across additional coral species configurations and across broader spatial scales is now needed. Our study coincided with a large-scale bleaching event [44], resulting in rapid coral loss and changes in reef fish assemblage structure among our study sites [6], and likely affected the foraging behaviour of a range of reef fish species including herbivores [15,65,75]. Although the present study provides clear evidence of how herbivory processes can vary with coral species composition, it was carried out in the context of this disturbance. Disturbance dynamics are complex [15,70], and it is likely that fish assemblages are in transition with changes in coral cover. Further research into the spatio-temporal variation in foraging behaviour of individuals and functional groups across such disturbances would improve our understanding of how changing reef configurations interact with climate change impacts to influence critical ecological functions [15,16].

Understanding causal links between habitat species composition and ecosystem function is of growing concern in this era of unprecedented and rapid ecological change [5,7,9]. In particular, elucidating how the increasing modification of ecological communities affects ecosystem processes is central to our capacity to anticipate whether new species configurations will continue to provide goods and services as required by societies that depend on them [14,17,23]. On coral reefs, whether herbivores can compensate for increased algal production as coral cover decreases, and maintain critical rates of algal consumption will be fundamental to the persistence of reconfigured coral-dominated systems [66]. Our results show that herbivore assemblage structure varied among the studied habitats, however, did not reflect the observed variation in herbivory rates. While grazing was insensitive to variation in coral composition, browsing varied considerably, indicating that different mechanisms determined by specific habitat characteristics may be shaping these key processes. While the precise mechanisms are not known, variation in browsing was best predicted by the composition and cover of benthic communities, and conversely the cover of dead substrata and macroalgae, characteristics that underscore the structural complexity of reef habitats and which may have influenced differential foraging behaviour. With ongoing degradation of coral reefs and the homogenization of both coral and fish assemblages [6,20], these results suggest that, within reefs, key ecosystem functions will likely vary among altered coral configurations, according to the differential vulnerability of corals to disturbances and ecological interactions between reef fishes and their environment [15]. More generally, our results emphasize the role of differential habitat characteristics and provide explicit

support for assigning greater concern to the composition and structure—as well as cover—of habitat-building species in assessments and management of ecosystem function [7,23].

Data accessibility. All primary data are archived and accessible at the GitHub repository: https://github.com/LauraERichardson/Herbivory [76].

Authors' contributions. L.E.R., N.A.J.G. and A.S.H. conceived the ideas and designed methodology, L.E.R. collected and analysed the data, L.E.R. led the writing of the manuscript. All authors contributed critically to the drafts and gave final approval for publication.

Competing interests. We declare we have no competing interests.

Funding. This study was funded by the Australian Research Council to A.S.H. (DE130100688).

Acknowledgements. We thank Brock Bergseth, Jacob Eurich, Alexia Graba-Landry, Molly Scott and Lizard Island Research Station staff for field support, Murray Logan and Rie Hagihara for statistical advice; and six anonymous reviewers for their helpful comments.

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
