## [Reviewer comments · Proceedings of the Royal Society B: Biological Sciences]

Review History

RSPB-2019-2214.R0 (Original submission)

Review form: Reviewer 1

Recommendation

Accept with minor revision (please list in comments)

Scientific importance: Is the manuscript an original and important contribution to its field?

Excellent

General interest: Is the paper of sufficient general interest?

Good

Quality of the paper: Is the overall quality of the paper suitable?

Excellent

Is the length of the paper justified?

Yes

Should the paper be seen by a specialist statistical reviewer?

No

Do you have any concerns about statistical analyses in this paper? If so, please specify them explicitly in your report.

No

It is a condition of publication that authors make their supporting data, code and materials available - either as supplementary material or hosted in an external repository. Please rate, if applicable, the supporting data on the following criteria.

Is it accessible?

No

Is it clear?

N/A

Is it adequate?

N/A

Do you have any ethical concerns with this paper?

No

Comments to the Author

Overall, this is a very well written manuscript that is clear and engaging. The authors did a thorough and thoughtful job of responding to previous reviewers, and as a result, the project is described in a straightforward manner that is easy to follow. This work will definitely be of interest to coral reef researchers, particularly those studying herbivory, but I also expect that it will find an audience in the broader scientific community. The authors discuss the preferential use of grasslands by some terrestrial herbivores in the Discussion (Line 389) and contextualize the results in a larger ecological framework (Lines 465-470). With the inclusion of a couple additional comparisons to systems beyond coral reefs the authors could address an expanded audience.

My specific comments are as follows:

Lines 23-4: In the abstract the authors state that one of their main aims is to examine “how herbivorous fish assemblages... differ among three taxonomically distinct, replicated coral habitats” but do not address this in the discussion. This aim is seemingly reiterated in Line 112. It seems, however, that the primary objective as stated in Lines 107-108 is more accurately reflected in the discussion of this paper. I would recommend that the authors either revise the wording of their aims elsewhere in the manuscript to reflect Lines 107-108, or include a discussion of the differences in herbivore assemblages between the distinct coral habitats. This is addressed to some extent in Figures 1-2 and Supplemental Tables 1,3, and 4, but if it is a primary question, it should be included in the text of the Discussion.

Line 112: “Does” should be “do” in this sentence.

Line 171: Can the use of 8 vs. 9 (and later 2 vs. 3 in Line 178) be clarified here?

Lines 277-278: It was surprising to see that while the total coral cover was significantly different between site types, the total cover of bleached coral was not, but the reasoning for this (Lines 405-406) was explained well in the Discussion.

Line 319: Misplaced comma

Lines 423-425: Could the authors clarify the wording in this sentence? Is the issue that only those fish that consume brown macroalgae are being classified as browsers? If so, that seems to be an issue in the classification scheme rather than an issue with visual estimates.

Tables S7 and S8: These are not referred to in the manuscript text and do not have any explanation in the Supplemental Information. Table S7 is somewhat confusing and could likely be omitted.

Review form: Reviewer 2

Recommendation

Accept as is

Scientific importance: Is the manuscript an original and important contribution to its field?

Good

General interest: Is the paper of sufficient general interest?

Marginal

Quality of the paper: Is the overall quality of the paper suitable?

Good

Is the length of the paper justified?

Yes

Should the paper be seen by a specialist statistical reviewer?

No

Do you have any concerns about statistical analyses in this paper? If so, please specify them explicitly in your report.

No

It is a condition of publication that authors make their supporting data, code and materials available - either as supplementary material or hosted in an external repository. Please rate, if applicable, the supporting data on the following criteria.

Is it accessible?

Yes

Is it clear?

Yes

Is it adequate?

Yes

Do you have any ethical concerns with this paper?

No

Comments to the Author

In my view, the authors have constructively addressed the concerns raised during the initial review. Furthermore, as a new reviewer, I found the revised manuscript to be conceptually and technically sound.

I do generally question whether the manuscript will be of interest to readers outside of the "coral

reef world", but if the other reviewers and editorial team deem this manuscript to be broadly interesting, then I think it is ready for publication.

Decision letter (RSPB-2019-2214.R0)

26-Nov-2019

Dear Dr Richardson:

Your manuscript has now been peer reviewed and the reviews have been assessed by an Associate Editor. The reviewers' comments (not including confidential comments to the Editor) and the comments from the Associate Editor are included at the end of this email for your reference. As you will see, the reviewers and the Editors have raised some concerns with your manuscript and we would like to invite you to revise your manuscript to address them.

Research ethics:

Use of animals and field studies:

Please submit a copy of your revised paper within three weeks. If we do not hear from you within this time your manuscript will be rejected. If you are unable to meet this deadline please let us know as soon as possible, as we may be able to grant a short extension.

Best wishes,
Dr Daniel Costa
mailto:proceedingsb@royalsociety.org

Reviewer(s)' Comments to Author:

Referee: 1

Comments to the Author(s)

Overall, this is a very well written manuscript that is clear and engaging. The authors did a thorough and thoughtful job of responding to previous reviewers, and as a result, the project is described in a straightforward manner that is easy to follow. This work will definitely be of interest to coral reef researchers, particularly those studying herbivory, but I also expect that it will find an audience in the broader scientific community. The authors discuss the preferential use of grasslands by some terrestrial herbivores in the Discussion (Line 389) and contextualize the

results in a larger ecological framework (Lines 465-470). With the inclusion of a couple additional comparisons to systems beyond coral reefs the authors could address an expanded audience.

My specific comments are as follows:

Lines 23-4: In the abstract the authors state that one of their main aims is to examine “how herbivorous fish assemblages... differ among three taxonomically distinct, replicated coral habitats” but do not address this in the discussion. This aim is seemingly reiterated in Line 112. It seems, however, that the primary objective as stated in Lines 107-108 is more accurately reflected in the discussion of this paper. I would recommend that the authors either revise the wording of their aims elsewhere in the manuscript to reflect Lines 107-108, or include a discussion of the differences in herbivore assemblages between the distinct coral habitats. This is addressed to some extent in Figures 1-2 and Supplemental Tables 1,3, and 4, but if it is a primary question, it should be included in the text of the Discussion.

Line 112: “Does” should be “do” in this sentence.

Line 171: Can the use of 8 vs. 9 (and later 2 vs. 3 in Line 178) be clarified here?

Lines 277-278: It was surprising to see that while the total coral cover was significantly different between site types, the total cover of bleached coral was not, but the reasoning for this (Lines 405-406) was explained well in the Discussion.

Line 319: Misplaced comma

Lines 423-425: Could the authors clarify the wording in this sentence? Is the issue that only those fish that consume brown macroalgae are being classified as browsers? If so, that seems to be an issue in the classification scheme rather than an issue with visual estimates.

Tables S7 and S8: These are not referred to in the manuscript text and do not have any explanation in the Supplemental Information. Table S7 is somewhat confusing and could likely be omitted.

Referee: 2

Comments to the Author(s)

In my view, the authors have constructively addressed the concerns raised during the initial review. Furthermore, as a new reviewer, I found the revised manuscript to be conceptually and technically sound.

I do generally question whether the manuscript will be of interest to readers outside of the "coral reef world", but if the other reviewers and editorial team deem this manuscript to be broadly interesting, then I think it is ready for publication.

Author's Response to Decision Letter for (RSPB-2019-2214.R0)

See Appendix A.

RSPB-2019-2214.R1 (Revision)

Review form: Reviewer 1

Recommendation

Accept as is

Scientific importance: Is the manuscript an original and important contribution to its field?

Good

General interest: Is the paper of sufficient general interest?

Good

Quality of the paper: Is the overall quality of the paper suitable?

Good

Is the length of the paper justified?

Yes

Should the paper be seen by a specialist statistical reviewer?

No

Do you have any concerns about statistical analyses in this paper? If so, please specify them explicitly in your report.

No

It is a condition of publication that authors make their supporting data, code and materials available - either as supplementary material or hosted in an external repository. Please rate, if applicable, the supporting data on the following criteria.

Is it accessible?

Yes

Is it clear?

Yes

Is it adequate?

Yes

Do you have any ethical concerns with this paper?

No

Comments to the Author

The authors have constructively addressed my concerns. I believe the manuscript is now ready for publication.

Review form: Reviewer 2

Recommendation

Accept as is

Scientific importance: Is the manuscript an original and important contribution to its field?
Excellent

General interest: Is the paper of sufficient general interest?
Excellent

Quality of the paper: Is the overall quality of the paper suitable?
Excellent

Is the length of the paper justified?
Yes

Should the paper be seen by a specialist statistical reviewer?
No

Do you have any concerns about statistical analyses in this paper? If so, please specify them explicitly in your report.
No

It is a condition of publication that authors make their supporting data, code and materials available - either as supplementary material or hosted in an external repository. Please rate, if applicable, the supporting data on the following criteria.

Is it accessible?
Yes

Is it clear?
Yes

Is it adequate?
Yes

Do you have any ethical concerns with this paper?
No

Comments to the Author
The authors addressed all concerns raised. In my opinion, the paper is ready for publication.

Decision letter (RSPB-2019-2214.R1)

27-Jan-2020

Dear Dr Richardson

I am pleased to inform you that your manuscript entitled "Coral species composition drives key ecosystem function on coral reefs" has been accepted for publication in Proceedings B.

Open Access

Paper charges

Sincerely,

Dr Daniel Costa

Appendix A

Reviewer(s)' Comments to Author:

Referee: 1

Comments to the Author(s)

Overall, this is a very well written manuscript that is clear and engaging. The authors did a thorough and thoughtful job of responding to previous reviewers, and as a result, the project is described in a straightforward manner that is easy to follow. This work will definitely be of interest to coral reef researchers, particularly those studying herbivory, but I also expect that it will find an audience in the broader scientific community. The authors discuss the preferential use of grasslands by some terrestrial herbivores in the Discussion (Line 389) and contextualize the results in a larger ecological framework (Lines 465-470). With the inclusion of a couple additional comparisons to systems beyond coral reefs the authors could address an expanded audience.

We are pleased that the referee found the paper clear, engaging, and that we had thoroughly responded to the previous reviewer comments. As recommended, to ensure the study reaches an expanded audience we have now included additional comparisons to non-reef systems including alpine forests (Ferrari et al. 2009), temperate intertidal rocky shores (Metcalf 1984), European grasslands (Fernández-Juricic et al. 2005), further evidence from African savannahs (Underwood 1982), and temperate arable areas (Whittingham et al. 2004) in Lines 368-375. We also relate the study findings to foraging seabirds of temperate rocky coastal systems (Wilson et al. 2011), temperate marine systems (Holbrook and Schmitt 1988) (Lines 386-388), and terrestrial processes of dung decomposition (Rosenlew and Roslin 2008), and pollination by bees (Larsen et al. 2005) (Lines 402-405).

My specific comments are as follows:

Lines 23-4: In the abstract the authors state that one of their main aims is to examine “how herbivorous fish assemblages... differ among three taxonomically distinct, replicated coral habitats” but do not address this in the discussion. This aim is seemingly reiterated in Line 112. It seems, however, that the primary objective as stated in Lines 107-108 is more accurately reflected in the discussion of this paper. I would recommend that the authors either revise the wording of their aims elsewhere in the manuscript to reflect Lines 107-108, or include a discussion of the differences in herbivore assemblages between the distinct coral habitats. This is addressed to some extent in Figures 1-2 and Supplemental Tables 1,3, and 4, but if it is a primary question, it should be included in the text of the Discussion.

Thanks for highlighting this. Assessing variation in herbivore fish assemblage structure among distinct coral habitats, as well as grazing and browsing functions, was indeed a primary objective of the study in order to elucidate causal links between reef species composition and ecosystem functions. We agree with the referee that further discussion was needed to better address the variation found in herbivore assemblages beyond the existing descriptions in the discussion (L334-336, L358-361, L391-394, L401-402). To this end, we have added further detail highlighting how the composition of herbivorous fish assemblages differed among habitats in the results (mainly grazing species; L286-292), and more explicitly described these among-habitat differences in the discussion (L416-418, L463-465).

Line 112: “Does” should be “do” in this sentence.

Corrected, thanks.

Line 171: Can the use of 8 vs. 9 (and later 2 vs. 3 in Line 178) be clarified here?

Apologies for the lack of clarity here. Unfortunately, logistic constraints and the loss of some tiles precluded a completely balanced experimental design (we had 79 tiles to use across the nine sites). As such we deployed a minimum of eight tiles with established algal turf communities at each site (six were exposed to local herbivores and two placed within herbivore exclusions cages). The remaining seven tiles were used as additional caged-controls at seven out of the nine sites (all except one mixed coral site and one soft coral site). We have now clarified this in the methods on Lines 166-183.

Lines 277-278: It was surprising to see that while the total coral cover was significantly different between site types, the total cover of bleached coral was not, but the reasoning for this (Lines 405-406) was explained well in the Discussion.

We are glad the referee found our explanation for the among habitat variation in coral cover vs bleaching to be well explained.

Line 319: Misplaced comma.

Removed, thanks.

Lines 423-425: Could the authors clarify the wording in this sentence? Is the issue that only those fish that consume brown macroalgae are being classified as browsers? If so, that seems to be an issue in the classification scheme rather than an issue with visual estimates.

Apologies for the confusion here, and we thank the reviewer for raising this point. It is indeed an issue in the classification scheme rather than the visual estimates. In our study, we classified all species known to feed on fleshy macroalgae as browsers, not only those that consume brown macroalgae. However, we intended to highlight that there is a potential bias in the existing literature towards studies classifying browsers as those feeding on fleshy brown macroalgae such as *Sargassum* spp. We have now clarified this in the discussion on Lines 408-412.

Tables S7 and S8: These are not referred to in the manuscript text and do not have any explanation in the Supplemental Information. Table S7 is somewhat confusing and could likely be omitted.

Thanks for highlighting this. We have added references to Table S8 (now Table S4) and removed Table S7 as suggested from the Supplemental Information.

Referee: 2

Comments to the Author(s)

In my view, the authors have constructively addressed the concerns raised during the initial review. Furthermore, as a new reviewer, I found the revised manuscript to be conceptually and technically sound.

I do generally question whether the manuscript will be of interest to readers outside of the "coral reef world", but if the other reviewers and editorial team deem this manuscript to be broadly interesting, then I think it is ready for publication.

We are glad that Referee 2 considers the manuscript to be both conceptually and technically sound, and ready for publication. Addressing the point also raised by Referee 1, we have included additional comparisons to other ecological systems in order to address an expanded audience, as outlined above.